# Depressive Symptoms in Swiss University Students during the COVID-19 Pandemic and Their Correlates

**DOI:** 10.3390/ijerph18041458

**Published:** 2021-02-04

**Authors:** Thomas Volken, Annina Zysset, Simone Amendola, Anthony Klein Swormink, Marion Huber, Agnes von Wyl, Julia Dratva

**Affiliations:** 1Department of Health, Institute of Health Sciences, Zurich University of Applied Sciences, 8400 Winterthur, Switzerland; annina.zysset@zhaw.ch (A.Z.); kleinant@students.zhaw.ch (A.K.S.); marion.huber@zhaw.ch (M.H.); julia.dratva@zhaw.ch (J.D.); 2Department of Dynamic and Clinical Psychology, Faculty of Medicine and Psychology, Sapienza University of Rome, 00185 Rome, Italy; simone.amendola@uniroma1.it; 3School of Applied Psychology, Zurich University of Applied Sciences, 8005 Zurich, Switzerland; agnes.vonwyl@zhaw.ch; 4Medical Faculty, University of Basel, 4001 Basel, Switzerland

**Keywords:** depression, COVID-19, PHQ-9, students, pandemic, mental health, health survey, young adult

## Abstract

Background: COVID-19 containment measures and the uncertainties associated with the pandemic may have contributed to changes in mental health risks and mental health problems in university students. Due to the high burden of the disease, depression is of particular concern. However, knowledge about the prevalence of depressive symptoms in Swiss university students during the pandemic is limited. We therefore assessed the prevalence of depressive symptoms and their change during the COVID-19 pandemic in a large sample of Swiss university students. Methods: We assessed depressive symptoms in two cross-sectional cohorts of university students (*n* = 3571) in spring and autumn 2020 during the COVID-19 pandemic and compared them with a matched sample of the Swiss national population (*n* = 2328). Binary logistic regression models estimated prevalence with corresponding 95% confidence intervals (95% CI). Results: Adjusted prevalence of depressive symptoms in female (30.8% (95% CI: 28.6–33.0)) and male students (24.8% (95% CI: 21.7–28.1)) was substantially higher than in the matching female (10.9% (95% CI: 8.9–13.2)) and male (8.5% (6.6–11.0)) pre-pandemic national population. Depressive symptoms in the two consecutive student cohorts did not significantly differ. Conclusions: More than a quarter of Swiss university students reported depressive symptoms during the COVID-19 pandemic, which was substantially higher as compared to the matched general population. Universities should introduce measures to support students in such times of crisis and gain an understanding of the factors impacting mental health positively or negatively and related to university structures and procedures.

## 1. Introduction

In March 2020, the Swiss government took drastic measures to contain the spread of the SARS-CoV-2 infections and protect the population. Classroom teaching, non-essential retail commercial activities, and gatherings of more than five people were banned and borders were partially closed [1,2]. From the end of April to the end of September, the restrictions were successively relaxed, and from June onwards, most public and private activities were permitted again under certain conditions, such as compulsory wearing of masks and compliance with minimum distance rules. While classroom teaching in schools was in principle possible again at an early stage, most universities decided to continue online teaching until the end of the spring semester and only switched back to classroom teaching and hybrid teaching forms from the autumn semester onwards. 

The measures taken to contain the spread of the coronavirus, such as social distancing and quarantine, can potentially cause negative psychological effects and increasing feelings of isolation. More specifically, fear of infecting oneself or others, frustration, boredom, interruption of professional activities, and financial loss could have an impact on mental health and well-being [3]. There is a growing body of published studies exploring symptoms of psychological distress due to the COVID-19 pandemic. During the initial outbreak in China, 8%, 29%, and 16% of the general population reported moderate-to-severe stress, anxiety, and depression, respectively, with no significant reduction in scores between the baseline and a four-week follow-up [4]. Similarly, a study from the UK found that 24% and 31% of the adult population experienced moderate-to-severe anxiety and depressive symptoms, respectively, with no or little improvement at an eight-week follow-up [5]. 

However, the pattern of mental health across the lifespan has been found to be highly dependent on the national context [6]. In Switzerland, the evidence for depression is mixed. While the age group of 15–34 years had a substantially higher prevalence of mild and moderate depressive symptoms than older age groups, the prevalence of diagnosed depression was much lower [7]. Accordingly, the COVID-19 containment measures and the uncertainties associated with the pandemic may have contributed to a further increase in mental health risks and mental health problems in university students transitioning through this vulnerable life phase.

From a mental health perspective, emerging adulthood—which includes the university years—has been considered a life course stage of elevated risk due to increased responsibilities and demands for self-direction [8,9,10], which contributes to changes in health or risk-taking and/or health-promoting behaviors [11]. Furthermore, emerging adulthood has been found to be a period where mental health problems peak [12,13,14,15], the prevalence of stress is high [16], and sleep disorders are common, the latter being a major risk factor for depression [17]. 

In accordance with the biopsychosocial model of depression [18], young adults and college students may experience greater uncertainty about their academic and professional career, as well as financial and health concerns for loved ones, that could lead to increased depressive symptoms. This assumption has also been supported by Keller and Nesse’s research [19], having investigated the situation–symptom congruence hypothesis and showing that different adverse and stressful situations can potentially lead to different depressive symptom patterns serving distinguishable adaptive functions. In particular, social losses (i.e., death, romantic loss, social isolation) were related more to symptoms than failures [19]. Among the Darwinian models of depression [20], the attachment theory offers a framework to understand the emergence of depression and depressive symptoms. During stressful situations, individuals with insecure attachment are more prone to fail in regulating emotions and experience symptoms of depression, as a way of signaling and maximizing their attachment needs in order to receive more support and protection [21].

Due to the high burden of the disease, depression is of particular concern [13,14,22]. The annual economic burden of depression in Switzerland has been estimated to be about 8 billion Swiss francs [23]. During the COVID-19 pandemic, a consistently high prevalence of depressive symptoms was found in Chinese (23.3%) [24] and Ukrainian students (31.7%) [25], while other studies reported an even higher prevalence of depressive symptoms in French (43.0%) [26] and Bangladeshi (53.7%) [27] university students. However, very few studies so far have assessed change in depressive symptoms among students over time. A large study from the United States found that the prevalence of depressive symptoms among students more than doubled between March/July 2019 and May/July 2020 (32%) [28]. A study of 200 university students in Switzerland reported a within-student mean increase of 4.4 points on the Center for Epidemiologic Studies Depression scale between September 2019 and April 2020, but did not find significant differences of depression scores in comparison to a cohort of students assed a year earlier (April 2019) of the same study major [29].

To date, very few cohort or repeated cross-sectional studies exploring change in depressive symptoms among university students during the COVID-19 pandemic have been published, and to the best of our knowledge, no study has so far assessed the respective change among Swiss university students. Moreover, the current knowledge about the prevalence of depressive symptoms in Swiss university students during the pandemic is limited due to study characteristics such as small sample size. Given these limitations, the present study´s aims were (1) to assess the prevalence of depressive symptoms and their change during the COVID-19 pandemic in a large sample of Swiss university students, (2) to compare the prevalence of depressive symptoms among students during the pandemic with the respective prevalence of a matched sample of young adults from the general population before the pandemic, and (3) to identify and explore risk and protective factors associated with depressive symptoms among students during the pandemic. Based on previous studies and in line with the theoretical framework outlined above, we hypothesized that female gender [24,25,26,27,30,31,32,33,34], low social status [24], foreign nationality [33], poor social support [33], low resilience [24,35], low self-efficacy [36,37,38,39], COVID-19-related concerns (health, economic situation) [26,33], and health risk behavior (binge drinking, cannabis consumption, lack of physical activity) [25,27,32,33] would increase the likelihood of reporting depressive symptoms. 

## 2. Materials and Methods

### 2.1. Study Design

The study was a comparative analysis of a population-based, cross-sectional health survey (2017) and a repeated cross-sectional health survey of students of a Swiss university of applied sciences during the COVID-19 pandemic in spring and autumn 2020.

### 2.2. Study Population and Data

The Swiss Health Survey (SHS) undertaken by the Swiss Federal Statistical Office (SFSO) is a nationwide survey on health status, health service utilization, and health-related behavior. The SHS employs telephone interviews and subsequent written questionnaires; it was first conducted in 1992 and is repeated every five years. For each survey year, a multistage probability sample is drawn of the permanent resident (including foreign nationals) population in Switzerland after stratification by the three predominant language/geographic regions (German, French, and Italian). Samples include individuals aged 15 years or older living in private households. Data were collected and administered by the SFSO under the regulation of the Federal Statistics Act (FSA) of 1992, which is a framework of law dedicated to federal data collection, data protection, and data security. Participants provide informed consent, which accommodates all future use of the data for research (FSA, 1992). For our study, we obtained the most recent SHS data (2017). The respective net sample size comprised *N* = 22,134 respondents, representing 7,036,199 subjects in the general population. 

Data for university students were derived from the “Health in Students study during the Corona pandemic” (HES-C), which aims to (1) evaluate the health of students during the pandemic, (2) investigate changes in health behavior and associated factors, as well as (3) assess student’s perception of the pandemic and related measures and their impact on students’ lives. All enrolled students of the Zurich University of Applied Sciences (ZHAW) (*N* = 13,500) were invited to participate in four consecutive web surveys that were administered between April and October 2020. In the present study, we used the pooled cross-sectional data from the first (*n* = 2363) and fourth (*n* = 1208) survey waves. Each survey lasted about 20–25 minutes and ran for a total period of seven working days (i.e., from April 3rd to 14th 2020 for the first wave and from October 5th to 13th 2020 for the fourth wave). Participants’ informed consent was obtained before starting the survey. The study was approved by both the local cantonal ethics committee (BASEC-Nr. Req-2020-00366) and the ZHAW data protection officer. 

For the comparison between the general Swiss population and university students, SHS and HES-C data were pooled, yielding an initial combined sample size of *n* = 25,705. Moreover, we extracted a demographically stratified sample of female and male participants aged 21–30 years who had completed secondary or tertiary education (*n* = 2028) from the SHS to match the corresponding students’ 10–90th percentile age range (*n* = 2328), yielding a restricted pooled sample of *n* = 4356. While the SHS participants in the restricted sample were very similar to the university students, certain differences with respect to education and other characteristics may exist nonetheless. SHS respondents indicated their education level, but we lacked information on whether they were completing a higher level of education at that time; i.e., respondents who had completed secondary education may or may not have been studying at the tertiary level at the time the SHS was administered. 

However, based on the Swiss Federal Survey on Adolescents 2010/2011 (ch-x), which collected data from 10,740 male subjects during the recruitment of compulsory military service, Barth and colleagues [15] reported differences in depressive symptoms between young adults with mandatory education as compared to those in the highest education category (grammar school or higher), but not between the highest education category and young adults who completed vocational training. Given that this male cohort was between 18–25 years old, with a mean of 19.7 years and a standard deviation of 1.1 years, we can confidently assume that the vast majority of men in the highest education category were still studying at the time of recruitment, while those with vocational training and mandatory education were already engaged in paid work. The findings from the ch-x cohort clearly indicate that in Switzerland, there are hardly any differences in depressive symptoms between young adults with at least vocational training and those with further education [15]. Accordingly, we assumed that the young adults in the restricted SHS sample were sufficiently comparable to our students to be used as a contrast for comparing depressive symptoms in young adults before and during the pandemic. This was also confirmed by our sensitivity analyses, which showed high agreement in the prevalence of depressive symptoms among young men in the restricted SHS sample and the corresponding ch-x cohort. 

### 2.3. Measures

#### 2.3.1. Outcome Measure: Depressive Symptoms

We used the Patient Health Questionnaire (PHQ-9) to assess participants’ symptoms of depression during the last 14 days [40]. Our choice was based on the brevity of the instrument, its excellent psychometric properties [40], and its high usage in COVID-19-related health surveys, which facilitated the comparability of our results. The PHQ-9 uses nine items with DSM-V diagnostic criteria to assess depressive symptomatology on a 4-point Likert scale ranging from 0 (not at all) to 3 (nearly every day). The overall scale scores capture depression severity and are computed as the sum of all items, ranging from 0 to 27, with higher scores indicating a higher level of depressive symptoms. The following levels of depression severity according to the overall scale scores have been defined [40]: minimal (1–4), mild (5–9), moderate (10–14), moderately severe (15–19), and severe (20–27). PHQ-9 cut-off scores between 8 and 11 were found to have acceptable diagnostic properties for detecting major depressive disorder [41]. In the present study, we used the established cut-off of 10 or above [24,25,26,42] to classify participants as having depressive symptoms (0 = no, 1 = yes), being the primary outcome of the present study.

#### 2.3.2. Study Covariates

Socio-demographic covariates included participants’ age at the last birthday in complete years, gender (0 = men, 1 = women), nationality (0 = Swiss, 1 = double citizen, 2 = foreign national), and social status of parents at student age 16 years using a modified McArthur scale ranging from 1 (lowest) to 10 (highest) [43]. 

COVID-19-related covariates comprised the concerns students had about their own health, the health of family members (parents, siblings, grandparents, own child/child of partner, other relatives), and the financial situation of family members. They could answer the question “Are you concerned about your own health in the context of the pandemic?” with either “I have no concerns” (=1), “…some concerns” (=2), “…big concerns” (=3), and “not relevant” (=4). Similarly, concerns about family members were assessed. Responses were dichotomized as 0 (i.e., “I have no concerns”, “I have some concerns”, ‘not relevant”) or 1 (i.e., “I have big concerns”).

Psychosocial covariates included social support, resilience, and self-efficacy. We used The Oslo Social Support Scale (OSSS-3) [44,45] to assess respondents’ social support. The OSSS-3 is a short questionnaire to explore social support through three items about number of close confidants, sense of concern or interest from other people, and relationship to neighbors. We used established cut-off values of OSSS-3 sum scores [44,45] to classify respondents into individuals with strong (0: 12–14), poor (1: 3–8), and moderate (2: 9–11) social support. 

In order to assess resilience, we used the Brief Resilient Coping Scale (BRCS) [46], a brief self-report questionnaire that assesses resilient coping, conceptualized as the tendencies to cope with stress in a highly adaptive manner. It comprises four items (e.g., Item 4: “I actively look for ways to replace the losses I encounter in life”). Participants responded on a 5-point Likert scale (from 1 = “doesn’t describe me at all” to 5 = “describes me very well”). The total score has a range of 4–20, with higher scores indicating higher resilience. Respondents were classified into groups with high (0: 17–20), low (1: 4–13), and moderate (2: 9–11) resilience according to established cut-off values [46]. The German Version of the BRCS showed good psychometric properties [47]. Self-efficacy was assessed using the Allgemeine Selbstwirksamkeit Kurzskala (ASKU; General Self-Efficacy Short Scale) [48]. The ASKU comprises three items (“In difficult situations, I can rely on my abilities”, “I can cope with most problems on my own”, and “Generally, I can handle strenuous and complicated tasks well”) and participants were asked to respond on a five-point Likert scale (from 1 = “strongly disagree” to 5 = “strongly agree”). The ASKU scale scores ranges from 1 to 5 and represents the mean score over all three items, with higher scores indicating higher self-efficacy. 

Health risk behavior-related covariates comprised binge drinking, cannabis consumption, and lack of physical activity. Participants were asked how many times (if any) they had 5 or more alcoholic drinks on one occasion during the past 30 days [49]. Participants responded on a 6-point Likert scale ranging from 1 (“never”) to 6 (“10 or more times”). The answer was dichotomized (0 = no, 1 = yes) in order to capture whether respondents had at least one binge-drinking episode in the past 30 days. 

Similarly, cannabis consumption in the past 30 days was assessed. Participants responded on a 9-point Likert scale ranging from 1 (“I do not use it”) to 9 (“10 or more times”). The answer was dichotomized (0 = no, 1 = yes) in order to capture marijuana consumption on at least one occasion in the past 30 days. Physical activity was assessed using the Global Physical Activity Questionnaire (GPAQ) [50], which captures weekly vigorous and moderate physical activities as well as sedentary behavior during work, leisure time, and commuting. Participants were classified as physically active (0 = no, 1 = yes) if they met the World Health Organization recommendations on physical activity for health, i.e., they achieved at least 600 Metabolic Equivalent (MET) minutes during a typical week. 

### 2.4. Statistical Analyses

We used Stata Version 15.1 (StataCorp, College Station, TX, USA) for statistical analyses. Descriptive statistics (i.e., frequencies, prevalence, mean, standard deviation) were applied to evaluate the characteristics of the samples. We reported estimated prevalence of depressive symptoms in students, the national population of Switzerland, and an age- and gender-restricted sample of the national population matching the respective students. Logistic regression models were employed to estimate crude and adjusted prevalence with corresponding 95% confidence intervals (95% CI). Adjustment comprised age, gender, and education. Differences between the national population and students were assessed using design-based F-tests, which took into account the complex survey structure of the SHS. Single and multivariable logistic regressions with robust standard errors were used to assess associations between students’ depressive symptoms and socio-demographic, psychosocial, COVID-19-related, and health behavior factors. We reported odds ratios (OR) with corresponding 95% CI and *p*-values. 

For all logistic models, we visually checked whether independent variables were related linearly to the log odds, checked an adequate minimum of 10 cases with the least frequent outcome for each independent variable, and assessed multicollinearity using the variance inflation factor (VIF) in the multivariable model. Underlying assumptions of the logistic models were met in all models; mean VIF in the multivariable model was 1.09 with individual variables ranging from 1.02 to 1.26. Statistical significance was established at *p* < 0.05.

## 3. Results

### 3.1. Socio-Demographic Characteristics of Respondents

Sociodemographic and depressive symptom characteristics of the initial combined SHS and HES-C samples are presented in Table 1. Not surprisingly, university students differed substantially and significantly from the general population with respect to sociodemographics. University students were younger (aged 26.0 ± 5.5 years) than the general population (aged 47.6 ± 18.7 years), more likely to be female (69.5% vs 50.6%), and represented a homogenous group with respect to their highest attained level of education (100% had completed secondary education). HES-C students who participated in autumn were slightly younger (25.0 ± 5.4 years) than those who participated in spring (26.4 ± 5.6 years). Students in the spring and the autumn survey did not significantly differ in terms of social status, nationality, resilience, concerns about their own health, and cannabis use. However, students in the autumn cohort reported slightly lower self-efficacy and slightly lower social support, and health and financial worries about family members were more prevalent. At the same time, they more frequently reported binge-drinking events and insufficient physical activity.

### 3.2. Prevalence of Depressive Symptoms among University Students during the Pandemic and among the General Population before the Pandemic

Crude prevalence of depressive symptoms was 8.6% (95% CI: 8.1–9.1) in the Swiss population and 27.2% (25.7–28.9) in university students (*p* < 0.001). The corresponding mean PHQ-9 score was 4.0 ± 3.9 in the Swiss population and 7.4 ± 4.9 in university students (*p* < 0.001). Depressive symptoms in the HES-C spring and autumn cohorts (27.1% and 27.7%) did not significantly differ (*p* = 0.740). Similarly, mean PHQ-9 scores were 7.3 ± 4.7 and 7.6 ± 5.0 (*p* = 0.2505). In order to adjust for age, gender, and education, we restricted the student sample to students aged 21–30 years, all who had completed secondary education, and compared it with a corresponding restricted SHS sample in the same age group who had completed secondary or tertiary education (Table 2). The adjusted prevalence of depressive symptoms in female (30.8% (95% CI: 28.6–33.0)) and male students (24.8% (95% CI: 21.7–28.1) was substantially higher than in the matching female (10.9% (95% CI: 8.9–13.2)) and male (8.5% (6.6–11.0)) pre-pandemic restricted national population (*p* < 0.001). Similarly, mean PHQ-9 depression scores were 7.9 (95% CI: 7.7–8.1) in female and 6.9 (95% CI: 6.5–7.2) in male students. In both genders, PHQ-9 scores were 2.8 points higher (95% CI: 2.4–3.2 and 2.3–3.2, respectively) as compared to the corresponding national population (*p* < 0.001). 

### 3.3. Factors Associated with Depressive Symptoms among University Students 

The following analyses are based on the pooled cross-sectional data from the spring and autumn student surveys (HES-C). With respect to socio-demographics, women, double citizens, and foreign nationals as compared to Swiss nationals were more likely to exhibit depressive symptoms, while older students and students from families with higher social status were less likely to have depressive symptoms. With the exception of age, all associations were statistically significant in unadjusted (Model 1) and adjusted (Model 2) models (Table 3). 

Adjusted for all other variables, women were 1.48 (95% CI: 1.09–2.02) times more likely to have depressive symptoms compared to men, and double citizens and foreign nationals as compared to Swiss nationals were 1.48 (1.06–2.07) and 1.81 (1.09–3.02) times more likely to exhibit depressive symptoms, respectively. Moreover, students from families with higher social status were less likely to have depressive symptoms (OR = 0.91; 95% CI: 0.84–0.99).

Psychosocial factors were also positively and statistically significant associated with the presence of depressive symptoms, both in the unadjusted and adjusted models. Adjusted for all other variables, university students with high perceived self-efficacy (OR = 0.37; 95% CI: 0.28–0.48) were less likely and those with low resilience (OR = 2.28; 95% CI: 1.52–3.43) and poor social support (OR = 2.35; 95% CI: 1.49–3.70) were more likely to have depressive symptoms compared to students with high resilience and strong social support, respectively. Students with moderate resilience (*p* = 0.325) or moderate social support (*p* = 0.236) did not significantly differ from those with high resilience or strong social support, respectively. 

With respect to COVID-19-related factors, we found positive associations between depressive symptoms and health concerns as well as family financial concerns but not for health concerns about family members in all estimated models. In the adjusted models, students who were concerned about their own health were 1.53 times (95% CI: 1.17–2.00) more likely to have depressive symptoms as compared to students who were not concerned about their own health. Furthermore, those who were worried about the financial situation of their family were 1.36 times (95% CI: 1.02–1.79) more likely to have depressive symptoms than students who were not concerned about the financial situation of their family. 

Health risk behavior, i.e., binge drinking, cannabis consumption, and low levels of physical activity, was positively associated with depressive symptoms in the unadjusted models. In the adjusted models, students who reported binge-drinking events in the past 30 days (OR = 1.42; 95% CI: 1.07–1.88) and those who did not meet the WHO-recommended total physical activity level of 600 or more MET minutes per week (OR = 1.41; 95% CI: 1.05–1.91) were more likely to have depressive symptoms as compared to students without binge-drinking events and those who met the WHO recommendation for physical activity, respectively. Adjusted for all other variables, cannabis use in the past 30 days was not significantly associated with depressive symptoms (*p* = 0.673).

## 4. Discussion

### 4.1. Prevalence of Depressive Symptoms

Overall, we found a high prevalence of depressive symptoms in university students during the COVID-19 pandemic. The crude prevalence in our Swiss student population was 27.2% (95% CI: 25.6–28.9) and 22.5% (95% CI: 20.6–24.4) adjusted for all covariates, respectively. Based on PHQ-9, a similar high prevalence of depressive symptoms during the COVID-19 pandemic was found in Chinese (23.3%; 95% CI: 21.5–25.1) [24] and Ukrainian students (31.7%; 4.1% in the severe category) [25], while other studies reported an even higher prevalence of depressive symptoms in French (43.0%; 7.0% in the severe category) [26] and Bangladeshi (53.7; 10.7% in the severe category) university students [27]. Differences between studies may be due to differences in the sample gender composition—i.e., women tend to have a higher prevalence of depressive symptoms—cultural differences, differences in education systems, or the different impact of the pandemic on different countries and regions [34]. 

Moreover, we found a substantially higher prevalence of depressive symptoms in female (30.8% (95% CI: 28.6–33.0)) and male (24.8% (95% CI: 21.7–28.1)) students during the COVID-19 pandemic as compared to a matched sample of the Swiss national population before the pandemic, where the respective prevalence was 10.9% (95% CI: 8.9–13.2) lower in women and 8.5% (6.6–11.0) lower in men (*p* < 0.001). This finding is consistent with previous cohort and cross-sectional studies of university students, which reported that students became more depressed during the COVID-19 pandemic restrictions [28,30,51]. A large study from the United States found that the prevalence of depressive symptoms among students more than doubled between March/July 2019 and May/July 2020 (32%) [28], and a large Chinese study reported a prevalence of 30.1% and 14.5% for students who were under quarantine versus students who were not under quarantine, respectively [52]. Having matched for age and education, our matched national sample was not perfectly congruent with our student sample. Consequently, we cannot determine conclusively what proportion of the differences between the prevalence of depressive symptoms is attributable to period effects, i.e., a “COVID-19 effect”, and what proportion is attributable to remaining differences in the characteristics between the two study populations. However, reports of the Swiss Health Observatory (Obsan) found that the prevalence of depression in the general population was 50–75% higher in French- and Italian-speaking regions as compared to German-speaking regions [7,53]. Consequently, the difference in the prevalence of depressive symptoms among SHS participants (all language regions) and HES-C participants (German-speaking region) could be even larger. In any case, the prevalence of depressive symptoms during the pandemic is alarmingly high among students, and faculty members need to acknowledge that many students are struggling during the pandemic. This is even more indicative as the prevalence of depressive symptoms between the two cross-sectional survey waves did not decline, despite the Federal Council’s significantly relaxed containment measures during late spring and early summer. This could be due to the fact that only the structure of the stressors changed between the two survey waves, but not their overall level, and thus their detrimental effect on students’ mental health remained. Because of the lockdown, spring 2020 in Switzerland can be described as a phase of increased risks in terms of social isolation, feelings of loneliness, and insecurities about studying, while at the same time, due to the newness of the situation, students reported hardly any fears about the financial or health situation of their families. In autumn, the situation was just the opposite: while at the time of the survey, social contacts were possible again almost without restrictions, students were clearly more concerned about the health and financial situation of their families, and risk-taking behaviors such as binge-drinking events and lack of physical exercise, which may reflect increasing frustration about the persistence of the pandemic, had increased. 

The complex impact on students’ private and academic lives must be disentangled to understand the unique effects of educational activities and societal precautionary measures. Furthermore, previous studies have demonstrated the close relationship between psychological distress and poor academic performance [54,55,56,57] and between psychological distress and career outcomes [58]. This represents yet another reason why universities should take the current crisis very seriously and provide appropriate support for those affected.

### 4.2. Factors Associated with Depressive Symptoms among University Students

In the present study, we identified a number of risk factors associated with depressive symptoms. With very few exceptions, these factors were consistent with our hypotheses and the existing literature on quarantine and the recently published literature on COVID-19-related depressive symptoms among university students. 

In the present study, female gender was associated with a higher risk of reporting depressive symptoms, which is in line with a large body of previous findings [24,25,26,27,30,31,32,33,34]. Moreover, students with a migration background, i.e., double citizens and foreign nationals, and students from families with lower socioeconomic status were more likely to have depressive symptoms. Previous studies reported similar associations of depressive symptoms with foreign nationality [33] and socioeconomic status [15,24]. 

Students who were worried about their own health were more likely to have depressive symptoms. This finding is in line with Wathelet and colleagues´ assessment of health-related effects of COVID-19 on mental health in French university students [33]. However, worrying about the health of relatives was not associated with depressive symptoms in our study, while Wathelet et al. reported a higher risk for students who were worried about the health of relatives [33]. Still, students who were worried about the financial situation of their families were more often affected by depressive symptoms, which is in line with Essadek and colleagues’ study of French university students during the pandemic [26].

With respect to psychosocial factors, students with poor social support, low resilience, and low self-efficacy were more likely to report depressive symptoms. Previous studies of French and Chinese students and adolescents during the pandemic found similar associations between weak feelings of social integration and mental health symptoms [33], low levels of resilience [24,35], and negative coping [35], which were risk factors for depressive symptoms. Similarly, pre-pandemic studies of Iranian students, Chinese adolescents, and Chinese athletes showed that low self-efficacy was a risk factor for depressive symptoms [36,37,38].

Finally, students with low physical activity and at least one binge-drinking episode in the past 30 days were more likely to report depressive symptoms. This finding is consistent with studies of Bangladeshi, Chinese, French, and Ukrainian students during the pandemic, which found that depressive symptoms were more prevalent in students with low levels of physical activity [25,27,33] or unhealthy lifestyles [32]. It is important to note that the relation between substance abuse and depressive symptoms is potentially bi-directional; i.e., symptoms of depression can predict increased likelihood of developing an alcohol-related disorder, while alcohol problems can predict future depressive symptoms. Moreover, alcohol and drugs can be used to cope with psychological distress such as depressive symptoms to alleviate negative affect [59,60,61]. However, we found no association between cannabis consumption and depressive symptoms in our study population. Whether a student’s depressive symptoms increased alcohol consumption or vice versa cannot be determined in this analysis. 

### 4.3. Study Limitations

The findings of this study should be interpreted with the following limitations in mind. Firstly, the use of self-reported measures could have increased the risk of social desirability patterns, affecting the results. Taboos and social stigma around depression as well as health behavior may have led to lower prevalences of both depression and psychoactive substance use. On the other hand, students burdened by the pandemic measures may have felt more inclined to participate in the survey. Secondly, we assessed change in depressive symptoms among students using two different cross-sectional student cohorts in spring and autumn. The cross-sectional design precluded pre-post linkage to ascertain individual-level change. That said, it is noteworthy that the aggregate level of reported depression remained high against substantively distinct backdrops of epidemiological progression and consequential social precautions. While we lacked data on the prevalence of depressive symptoms in students before the pandemic, we were able to estimate the prevalence by deriving a matched sample of the general population (age, gender, education) from the SHS as a proxy data for comparison.

## 5. Conclusions

The prevalence of depressive symptoms among Swiss university students during the COVID-19 pandemic was high in the beginning of the Swiss lockdown in April 2020 and continued to be high despite relaxed containment measures. Universities and faculty should acknowledge that more than a quarter of their students show depressive symptoms and address mental health issues. Measures to support students in such times of crisis as well as preventive programs should be introduced, and universities should gain an understanding of factors potentially impacting mental health positively or negatively related to university structures and procedures. 

## Figures and Tables

**Table 1 ijerph-18-01458-t001:** Socio-demographic characteristics and prevalence of depressive symptoms in the Swiss Health Survey ^1^ (SHS) and the Health in Students study during the Corona pandemic (HES-C).

Variable	SHS	HES-C	*p*	HES-C Spring	HES-C Autumn	*p*
**PHQ-9** (**%**)			<0.001			0.760
Minimal (0–4)	65.5	31.1		31.5	30.3	
Mild (5–9)	25.9	41.6		41.4	42.1	
Moderate (10–14)	5.9	17.9		18.1	17.4	
Moderately severe (15–19)	1.9	7.1		6.9	7.7	
Severe (20–27)	0.8	2.2		2.1	2.6	
**Depressive symptoms** (**%**)			<0.001			0.740
PHQ-9 ≥ 10	8.6	27.2		27.1	27.7	
**Gender** (**%**)			<0.001			0.611
Women	50.6	69.5		69.8	69.0	
Men	49.4	30.5		30.2	31.0	
**Age group** (**%**)			<0.001			<0.001
<21 years	7.3	5.4		2.5	11.0	
21–30 years	14.8	80.4		81.9	77.5	
31–40 years	16.8	11.0		12.2	8.7	
41–50 years	17.5	2.8		3.0	2.6	
>50 years	43.6	0.4		0.5	0.3	
**Education** (**%**)			<0.001			1.000
Primary	18.7	0.0		0.0	0.0	
Secondary	47.5	100.0		100.0	100.0	
Tertiary	33.8	0.0		0.0	0.0	
**Social status** (**mean ± SD**)	-	5.64 ± 1.65		5.64 ± 1.64	5.64 ± 1.67	0.447
**Nationality** (**%**)	-					
Swiss	-	72.6		73.8	69.7	0.073
Double citizen	-	18.6		17.9	20.4	
Foreign nationality	-	8.8		8.3	9.9	
**Self-efficacy** (**mean ± SD**)	-	3.81 ± 0.61		3.83 ± 0.62	3.77 ± 0.61	0.006
**Resilience** (**%**)	-					0.170
Low	-	20.6		19.8	22.4	
Moderate	-	49.2		49.2	49.1	
High	-	30.2		31.0	28.5	
**Social support** (**%**)	-					0.003
Poor	-	16.7		15.3	19.6	
Moderate	-	59.9		60.0	59.9	
Strong	-	23.4		24.7	20.5	
**Health concerns** (**%**)	-					0.337
No	-	55.9		56.5	54.6	
Yes	-	44.1		43.5	45.4	
**Family health concerns** (**%**)	-					<0.001
No	-	8.1		4.9	15.3	
Yes	-	91.9		95.1	84.7	
**Family financial concerns** (**%**)	-					<0.001
No	-	40.7		38.1	46.6	
Yes	-	59.3		61.9	53.5	
**Binge drinking** (**%**)	-					<0.001
No	-	65.3		69.3	57.3	
Yes	-	34.8		30.7	42.7	
**Cannabis consumption** (**%**)	-					0.604
No	-	88.5		88.7	88.1	
Yes	-	11.5		11.3	11.9	
**Physical activity** (**%**)	-					
No	-	22.1		20.2	26.5	0.001
Yes	-	77.9		79.8	73.5	
Sample size	22,134	3571		2363	1208	

^1^ Percentages based on population-weighted data. Two-week prevalence for depressive symptoms and depression severity. *p*-values for comparison between SHS and HES-C from design-based F-test. Sources: Swiss Federal Statistical Office, Swiss Health Survey (SHS) 2017; Institute of Health Professions, Zurich University of Applied Sciences, HES-C 2020.

**Table 2 ijerph-18-01458-t002:** Adjusted prevalence of depressive symptoms in the Swiss Health Survey ^1^ (SHS) and the Health in Students study during the Corona pandemic (HES-C).

Title	Women 21–30		*p*	Men 21–30		*p*
**PHQ-9 category (%)**	**SHS 2017**	**HES-C 2020**	<0.001	**SHS 2017**	**HES-C 2020**	<0.001
Minimal (0–4)	51.8 (48.3–55.3)	26.2 (24.1–28.4)		66.3 (62.6–69.8)	37.4 (33.9–41.1)	
Mild (5–9)	37.3 (34.0–40.8)	43.0 (40.6–45.4)		25.2 (22.0–28.7)	37.8 (34.3–41.5)	
Moderate (10–14)	7.9 (6.3–9.8)	20.6 (18.7–22.6)		5.6 (4.1–7.7)	16.5 (13.9–19.4)	
Moderately severe (15–19)	2.3 (1.4–3.8)	7.8 (6.6–9.2)		2.2 (1.3–3.7)	6.3 (4.7–8.3)	
Severe (20–27)	0.7 (0.3–1.6)	2.4 (1.7–3.3)		0.7 (0.3–1.6)	2.0 (1.1–3.3)	
**Depressive symptoms** (**%**)			<0.001			<0.001
PHQ-9 ≥ 10	10.9 (8.9–13.2)	30.8 (28.6–33.0)		8.5 (6.6–11.0)	24.8 (21.7–28.1)	
Sample size	1135	1625		893	703	

Difference between university students and the matching national population were assessed using design-based F-tests. Percentages are based on population-weighted data. Columns show percentages with 95% CI in parentheses. ^1^ Sources: Swiss Federal Statistical Office, Swiss Health Survey (SHS) 2017; Institute of Health Professions, Zurich University of Applied Sciences, HES-C 2020.

**Table 3 ijerph-18-01458-t003:** Factors associated with depressive symptoms among university students.

Title	Model 1			Model 2		
Variable	OR	*p*	95% CI	OR	*p*	95% CI
Gender (Ref = Men)						
Women	1.24	0.021	1.03–1.49	1.48	0.012	1.09–2.02
Age (years)	0.97	<0.001	0.96–0.99	0.98	0.117	0.96–1.00
Social status	0.89	<0.001	0.85–0.94	0.91	0.036	0.84–0.99
Nationality (Ref = Swiss)						
Double citizen	1.37	0.003	1.11–1.69	1.48	0.020	1.06–2.07
Foreign nationality	1.39	0.022	1.05–1.85	1.81	0.022	1.09–3.02
Self-efficacy	0.28	<0.001	0.24–0.33	0.37	<0.001	0.28–0.48
Resilience (Ref = high [17,18,19,20])						
Low (4–13)	4.91	<0.001	3.86–6.25	2.28	<0.001	1.52–3.43
Moderate (14–16)	1.89	<0.001	1.53–2.35	1.18	0.325	0.85–1.62
Social support (Ref = strong [12,13,14])						
Poor (3–8)	4.35	<0.001	3.34–5.66	2.35	<0.001	1.49–3.70
Moderate (9–11)	1.55	<0.001	1.24–1.94	1.22	0.236	0.88–1.70
Health concerns (Ref = No)						
Yes	1.56	<0.001	1.33–1.84	1.53	0.002	1.17–2.00
Family health concerns (Ref = No)						
Yes	1.21	0.235	0.88 – 1.65	1.09	0.746	0.64–1.88
Family financial concerns (Ref = No)						
Yes	1.51	<0.001	1.28–1.80	1.36	0.033	1.02–1.79
Binge drinking (Ref = No)						
Yes	1.39	0.001	1.15–1.68	1.42	0.016	1.07–1.88
Cannabis consumption (Ref = No)						
Yes	1.29	0.040	1.01–1.65	0.92	0.673	0.62–1.36
Physical activity (Ref = Yes)						
No	1.46	0.002	1.16–1.85	1.41	0.024	1.05–1.91
Constant	0.38	<0.001	0.35–0.41	0.06	<0.001	0.03–0.11
Sample size	3412–3569			3209		

Logistic regressions with robust standard errors. Dependent variable depressive symptom: y = 1 for PHQ-9 ≥ 10 and y = 0 for PHQ-9 < 10. Model 1: univariate analyses. Model 2: all variables in model. OR: Odds ratio; 95% CI: 95% confidence interval; Ref: reference category. Source: Institute of Health Professions, Zurich University of Applied Sciences, HES-C 2020.

## Data Availability

The data presented in this study are available on request from the corresponding author. The data are not publicly available due to legal and privacy issues.

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
