# Peer review of "Depressive Symptoms in Swiss University Students during the COVID-19 Pandemic and Their Correlates"

_ijerph, 2021, doi:10.3390/ijerph18041458_

Round 1

Reviewer 1 Report

Dear Authors, thank you for the possibility of reading your interesting paper. The undertaken issue of depressive symptoms in students during COVID-19 pandemic is highly important and fits into the IJERP’s scope and aims. The paper refers to predictors of depressive symptoms measured in two cross-sectional large cohorts of Swiss university students in April and October 2020. Additionally, Authors compare results of Swiss students during pandemic to results of global Swiss population in the age of 21-30 collected in the pre-pandemic period (2017).

The strength of this project is tackling the topic of students’ metal health during the ongoing pandemic. Authors were able to gather responses in two waves, what enabled to monitor the dynamics of depression symptoms’ levels.  The paper is well structured and well written. 

Nevertheless, there are a few points that should be elaborated and one serious flaw regarding the study design.

  1. Introduction:
    1. Although the style of introduction is clear, persuasive and literature review is adequate – it is insufficient. Firstly, the description of students as a specific social group is needed. The introduction is structed showing the general population mental health issues during pandemic, then the meaning of emerging adulthood with a short explanation how it affects students’ mental health. Students differ from the general population in the same age, and Authors should explain it in more detailed manner.
    2. Secondly, students differ in substance use from general population in the same age group. Any of the chosen depressive symptoms’ predictors are described in the introduction. Authors have introduced: resilience, social support, health concerns, substance use, and physical activity but refers to all those variables in just one sentence [lines 109-114] which seems to be too simplistic. Therefore, the justification of creating a model including all those variables is unclear.
    3. The paper is dedicated to the issue of depression. Although the prevalence of depression symptoms is well written, there is a lack of explaining theoretical model and clarifying the psychological mechanism behind depression. There should be an additional paragraph explaining theoretical background of depression.  
  2. The main concern is the study design. Although two measurements are the strength of this study - relevant and important for in-depth research about depression symptoms in students, the comparison to the general sample before pandemic in not justified. One could make such a comparison between students and the general population of the same group during pandemic. The other solution would be to compare student sample during pandemic with student sample before pandemic. Proposed in this study comparison of student sample with the general population before pandemic is incorrect. The general sample, even relatable in the aspect of age, differs from student sample with regard to mental health, substance abuse etc. Therefore, comparing the general sample of young adults before pandemic and students during pandemic, adds nothing to our knowledge about mental health during pandemic. It would be obvious, if Authors would describe how the student group differs from the general population in the introduction. I do not agree with Authors, that they made comparison with “the respective prevalence before pandemic” [line 107]. My firm suggestion is to remove this part of research, as the comparison of two samples of students and logistic regression of depressive symptoms make strong enough case for this paper to be published. Also, then the content would congruent with the paper’s title. Under this condition the paper should be published.
  3. Point 2.4 Statistical analysis – the conditions for logistic regression should be added
  4. Point 3.1. Results – The other key concern is the lack of a proper description of the variables used in the study. Readers are not aware how many students with double citizenship took part in the study and the levels of social status, self-efficacy, social support, health concern, financial concern, binge drinking, cannabis consumption, and physical activity are unknown.  I do understand that in the Table 1 Authors have shown also the significance of differences in the first and second wave of the study, and it might not be the most important for the depressive symptoms’ predictors, but those variables should be described thoroughly.
  5. Point 3.2 It should be added to the section’s title ‘with regard to gender’  
  6. Tables- minor issues: ‘p’ should be small letter, 0.0000 should be rather <0.001, and everywhere four places after a dot should be changes for three places
    1. Table should be readable separately, therefore it should be added in the Table 2., that 21-30 refers to age, and there should be sample size in each table (not only Table 1)
  7. Point 3.3 It is not clear what the sample size was (all students/ 1st wave, 2nd wave?)
  8. Point 4. Discussion. This section is well structured, clear and concise. Nevertheless, the lack of significance difference between depressive symptoms’ levels in students between two measurement is not explained. This result is of high importance, therefore in-depth discussion should be made. The theoretical background related to depression might be helpful here.
  9. Considering the introduction content and discussion, even though there are relevant comparisons to other studies, the paper lacks exploring depression in any theoretical model.  

Reviewer 2 Report

Drawing on survey data from the general and university student populations in Switzerland, this study compared the prevalence of depressive symptoms across the two populations. The authors reported that the unadjusted and adjusted levels of depressive symptoms were higher among college students than those in the general population, due primarily to the COVID-19 pandemic. In the second phase of the analysis, the authors assessed the associations between sociodemographic characteristics, the COVID-19 related stressors (e.g., self-reported financial and health related stresses), psychological resources (e.g., social support), resilience, coping style (self-efficacy), substance use, and levels of physical activities and depressive symptoms indicated by the Patient Health Questionnaire (PHQ-9). While the study was well-designed and used high quality data, I am not sure if the authors successfully answered “the so what question.” In what follows, I include my observations and comments and hope that they will be helpful for the authors’ revision endeavors.

As stated above, what is missing from this study is the absence of a strong theoretical framework, either public health (e.g., infectious disease or the pandemic) or mental health, that can be used to guide the analysis. This is particularly true for the analysis of covariates. I suggest that the authors focus sharply on the pernicious effects of the COVID-19 related stressors on mental health (i.e., depressive symptoms) and how college students in Switzerland utilized their social and psychological resources to cope with these stressors.  

For the most part, the analysis was done competently. I make three observations. First, I would suggest that the author focus on the adjusted results (i.e., omit the unadjusted results to save space) in their report since a wide range of covariates were included in the analysis. Second, I would stay away from any mentioning of “trajectory” as the authors analyzed two cross-sectional cohorts in a short period of time. The absence of multiple points in time and the panel data was not conducive for this type of “trend” analysis. And finally, often times the authors used inappropriate expressions. Please see examples I included below.        

Change the first sentence in the abstract to: “From a mental health perspective, emerging adulthood has been considered a life course stage of elevated risk due to increased responsibilities and demands for self-direction, which contributesto changes in health or risk-taking and/or health-promoting behaviors.”

Note that trajectory does not go well with cross-sectional studies. Please consider changing the sentence in the abstract to: “In this cross-sectional study, we assessed depressive symptoms (Patient Health Questionnaire, PHQ-9) in two cross-sectional cohorts of university students (n=3,571) in fall and autumn spring and fall (?) 2020 during the COVID-19 pandemic and compared them with the 2017 Swiss national population aged ≥15 years living in private households (n=22,134).”

Please change “Primary endpoint: depressive symptoms” to “Outcome measure: depressive symptoms.”

As can be seen from the above, a thorough line-editing will help improve the quality of the manuscript.

Reviewer 3 Report

1 - Why did the authors use the PHQ-9? I believe that a much broader screening scale characterized by a much
wider reference range than the 2 weeks indicated by the PHQ-9 for
the evaluation of symptoms.
2- Why do the authors refer to the diagnostic criteria of the DSM-IV if
the DSM - V is in use in the scientific community?

3- It would be useful for statistical purposes to compare the data of
Swiss students with those obtained by administering the same tests
to an ad hoc control group (for example, workers of the same age range)
instead of using data from another interview from 2017 that it does not
take into account the same variables.
4- It would be useful for the reader to create sub-paragraphs for
reading the "materials and methods used".

Round 2

Reviewer 1 Report

Dear Authors,

Thank you for adjusting your paper. All of my suggestions have been met and I accept your reason for including pre-pandemic data.

I believe that this is a high quality important paper as the students’ mental health is at risk.

Author Response

Thank you very much for your extremely helpful and constructive comments and suggestions, which helped us to improve the manuscript substantially.

Reviewer 2 Report

The authors made minimal changes to the manuscript in terms of theoretical development or construction that can be used as a roadmap for the study. Though I am not happy about this effort, the authors seem to have enough significant or interesting findings to report. However, I do suggest that the authors consider to shorten the abstract. It has 419 words!

Author Response

Thank you for pointing this out. We really appreciate your extremely helpful and constructive suggestions. We agree that the abstract is excessively long and we shortened it according to your suggestion (234 words), line 14-31.

Reviewer 3 Report

I thank the Authors for having satisfied my suggestions with their answers and modifications.

Author Response

(The authors gave the same response as above.)
